# Application of Moringa Leaf Extract as a Seed Priming Agent Enhances Growth and Physiological Attributes of Rice Seedlings Cultivated under Water Deficit Regime

**DOI:** 10.3390/plants11030261

**Published:** 2022-01-19

**Authors:** Shahbaz Khan, Danish Ibrar, Saqib Bashir, Nabila Rashid, Zuhair Hasnain, Muhammad Nawaz, Abdullah Ahmed Al-Ghamdi, Mohamed S. Elshikh, Helena Dvořáčková, Jan Dvořáček

**Affiliations:** 1National Agricultural Research Centre, Islamabad 45500, Pakistan; danish.uaar@gmail.com; 2Department of Soil & Environmental Science, Ghazi University, Dera Ghazi Khan 32200, Pakistan; 3Department of Botany, University of Agriculture, Faisalabad 38040, Pakistan; rasheed.nabila@yahoo.com; 4Department of Agronomy, PMA-Shah Arid Agriculture University, Rawalpindi 46000, Pakistan; zuhair@uaar.edu.pk; 5Department of Agricultural Engineering, Khwaja Fareed University of Engineering and Information Technology, Rahim Yar Khan 64000, Pakistan; dmnawaz@kfueit.edu.pk; 6Department of Botany and Microbiology, College of Science, King Saud University, P.O. Box 2455, Riyadh 11451, Saudi Arabia; abdaalghamdi@ksu.edu.sa (A.A.A.-G.); melshikh@ksu.edu.sa (M.S.E.); 7Department of Agrochemistry, Soil Science, Microbiology and Plant Nutrition, Faculty of AgriSciences, Mendel University in Brno, Zemědělská 1, 613 00 Brno, Czech Republic; helenadvorackovaa@gmail.com; 8Pedologiejh, spol s.r.o, Podstránská 692/71, 627 00 Brno, Czech Republic; pudoznalec@gmail.com

**Keywords:** antioxidant, moringa, priming, rice seedlings, water deficit

## Abstract

Population growth, food shortages, climate change and water scarcity are some of the frightening challenges being confronted in today’s world. Water deficit or drought stress has been considered a severe limitation for the productivity of rice, a widely popular nutritive cereal crop and the staple food of a large portion of the population. A key stage in crop growth is seed emergence, which is mostly constrained by abiotic elements such as high temperatures, soil crusting and low water potential, which are responsible for poor stand establishment. Seed priming is a pre-sowing treatment of seeds that primes them to a physiological state that allows them to emerge more proficiently. The purpose of this study was to investigate the potential of leaf extracts from local and exotic moringa landraces as seed priming agents in rice cultivated under water deficit (75% field capacity) and control conditions (100% field capacity). Rice seeds were placed in an aerated solution of moringa leaf extract (MLE) at 3% from three obtained landraces (Faisalabad, Multan and an exotic landrace of India). The results obtained from the experimentation show that the water deficit regime adversely affected the studied indicators including emergence and growth attributes as well as physiological parameters. Among the priming agents, MLE from the Faisalabad landrace significantly improved the speed and spread of emergence of rice seedlings (time to start emergence at 23%, emergence index at 75%, mean emergence time at 3.58% and final emergence percentage at 46%). All the priming agents enhanced the growth, photosynthetic pigments, gas exchange parameters and antioxidant activities, particularly under the water deficit regime, but the maximum improvement was recorded by the MLE from the Faisalabad landrace. Therefore, the MLE of the Faisalabad landrace can be productively used to boost the seedling establishment and growth of rice grown under normal and water deficit conditions.

## 1. Introduction

The human population is rapidly increasing, and food production is augmenting. There is massive pressure on all the available natural resources to feed the 9.7 billion people on earth by 2050. In addition, water scarcity and the changing climate are also leading challenges being faced by agricultural scientists. At present, the most distressing hazard for agricultural productivity is drought stress [1]. Among the cereals in the world, rice stands in third position, with a production of 495.74 million tones and cultivation on 167 million hectares, after maize and wheat [2]. Globally, three billion people consume rice as a staple food because it provides 20% of energy in the diet, whereas 19% and 5% energy supplies are from wheat and maize, respectively [3,4]. Field crop species are frequently exposed to disparaging environmental circumstances, especially abiotic stresses, which limit the yield and productivity of field crops [5]. Among these stresses, drought or water deficit is a major stress responsible for the low productivity of rice crops worldwide [6,7]. Drought stress also induces a set of biochemical, physiological and morpho-anatomical changes in plants to improve plant water use efficiency by minimizing the loss of water through reduced transpiration. In Asia, drought stress is one of the major limiting factors affecting rice productivity, with 23 M ha subject to drought conditions [8]. Plants are exposed to drought conditions when the loss of water through transpiration is higher than the supply of water to the root zone [9]. The level of mutilation produced by drought is normally capricious because it is influenced by a number of elements, particularly evapotranspiration, rainfall patterns and the water holding capacity of the soil.

Kim et al. [10] stated that water deficit conditions significantly influenced the morphology and growth of plant roots, the main component for nutrient and water absorption from the rhizosphere. It also causes a reduction in photosynthetic pigments and the photosynthetic rate, reduces the intake of CO_2_ and relative water contents and damages the processes of cell elongation and division [11]. Darwish et al. [12] reported that the WRKY transcription factor group plays a significant role in stress signaling pathways. Expression analysis revealed that TaWRKY32 was mainly expressed when plants were subjected to stress conditions. According to Nawaz et al. [13], drought stress at the reproductive stage (grain filling stage) is very dangerous because it adversely influences the partitioning of assimilates from source to sink. Water deficit conditions also lower the duration and speed of grain filling, which ultimately decreases the economical yield. Denaxa et al. [14] reported that reactive oxygen species, produced under drought stress, are detoxified by the important roles of enzymatic antioxidants, specifically catalase, superoxide dismutase and peroxidase, and non-enzymatic antioxidants. Farooq et al. [15] stated that there are various mechanisms including activation of antioxidant defensive systems, reduction in stomatal conductance and accumulation of compatible solutes that support crop plants in surviving under water deficit circumstances. Drought stress or aerobic environments caused variation in some parameters such as fruit TSS and firmness, the contents of carotenoids, flower number, proteins and activities of the catalase, peroxidase and superoxide dismutase enzymes [16].

In the agricultural sector, proficient water administration drives the handling of all possible routes for water demand and supply management approaches [17,18,19]. In Pakistan, Qamar et al. [20] reported that the productivity of field crops was highly dependent upon the availability of inputs, particularly water, at the critical growth stage, as the crop yield was significantly reduced by water shortage. Application of conservation tillage, stress signaling elements, plant water extracts, osmo-protectants and synthetic and natural mulches and the growth of drought-tolerant varieties are various practices and approaches that can be used to mitigate the adverse impacts of drought stress [21,22,23]. Soil puddling is also considered favorable for rice but unfavorable for post-rice upland crops [24]. Mineral fertilization is responsible for increased crop growth of rice [25]. In recent years, it has been reported that plant water extracts have a significant impact in enhancing the productivity of field crops [26]. Cheng and Cheng [27] reported that various crop water extracts have been identified in different plant species that improve the development, growth and productivity of field crops, particularly when cultivated under unfavorable circumstances, by manipulating physiological and biochemical processes including stomatal conductance, signal transduction, phytohormone metabolism, absorption of water/nutrients, antioxidant defensive systems and photosynthesis. Among most of the naturally available plant growth stimulants, leaf extracts of the moringa tree are at the top of the scientific community’s interests, being a natural source of many antioxidants, mineral elements, vitamins and growth-promoting substances [26,28,29]. *Moringa oleifera* is also considered as a potential allelopathic crop due to the presence of several allelochemicals [30,31].

Seed priming can be particularly beneficial to resource-poor farmers working in low-input agricultural systems where yield potential is limited by intrinsically stressed agronomic environments [32]. Usage of moringa leaf extracts via seed priming has been proven to enhance the emergence and establishment of seedlings and improve crop development and growth, leading to increased yields in unfavorable and normal environmental circumstances [26,33,34]. The ability of rice plants to cope with drought is supported by several parameters, including the structure of the root organ for water absorption [35]. Sarwar et al. [36] also stated that farmers can enhance crop productivity with the addition of organic supplements. Hence, the current experiment was designed to investigate the growth-promoting potential of leaf extracts obtained from exotic and local landraces of moringa when applied as seed priming agents to rice grown both under water deficit and normal conditions. It is also necessary to highlight that, prior to this study, no experiment thus far has explored the comparative seed priming potential of moringa leaf extracts (MLEs) from exotic and local landraces for rice crops.

## 2. Materials and Methods

### 2.1. Experimental Particulars

This experiment commenced during the rice cultivation season of 2018 and was set to explore the seed priming potential of leaf extracts obtained from exotic and local moringa landraces on emergence, physiological and gas exchange attributes, and enzymatic activities of rice seedlings (*Oryza sativa* L. Cul. Basmati Pak) grown under normal and water stress environments. The experiment was run at a greenhouse of the Department of Agronomy, Ghazi University, Dera Ghazi Khan, Pakistan. The trial was conducted according to a completely randomized design (CRD).

### 2.2. Treatment Plan, Drought Imposition, Extract Preparation and Application

The following treatment plan was followed for studying the mentioned objective:

Factor A—Drought levels:

Control conditions (CC, plants irrigated continuously at 100% field capacity);Drought stress (DS, plants irrigated continuously at 75% field capacity).

Factor B—Priming treatments:

Control (no priming);Hydro priming (priming with water);Priming with a water extract from white seeded moringa leaves (local landrace, Faisalabad origin), FM;Priming with a water extract from black seeded moringa leaves (local landrace, DG Khan origin), DM;Priming with a water extract from PKM1 moringa leaves (exotic landrace, established at Faisalabad), EM.

Two levels of water treatment (control conditions and drought stress) were induced by maintaining the soil moisture contents at 100% and 75% for the control conditions and drought stress, respectively. Plants grown under control conditions were irrigated at field capacity (FC) and considered as controls. Drought stress was imposed by maintaining the soil moisture contents at 75% as compared to FC. These soil moisture contents were maintained up to harvesting of rice seedlings starting from sowing. In order to determine the water loss by transpiration, pots were weighed on a daily basis, and soil moisture contents were sustained by recompensing this water loss through the addition of tap water up to the initial weight of the respective pot. The gravimetric method described by Nachabe [37] was followed to determine FC.

Fresh leaves were collected from a well-established moringa plantation at the respective sites which were healthy, mature and disease free. After rinsing under tap water, leaves were placed in a deep freezer overnight. A locally designed and assembled machine was used for juice extraction from overnight frozen leaves [34]. A 3% solution (100 mL) of moringa leaf extract (3 mL) was prepared by adding distilled water (97 mL) after sieving the extract.

Regarding seed priming treatments, seeds were placed in an aerated solution of moringa leaf extracts at 3%, for eight hours. Ten seeds were sown in each earthen pot (45 and 30 cm in height and diameter, respectively), filled with growth media (sand, clay, silt and compost in the same proportions). Overall, there were 80 pots in total: 40 for CC, and 40 for DS, as each priming treatment had four replications. Thinning was performed after fifteen days of emergence to maintain the four seedlings in each pot for further experimentation.

### 2.3. Emergence and Vigor Evaluation of Seedlings

ISTA protocols [38] were followed to record data of seedling vigor and emergence parameters. On a daily basis, emerged seedlings were counted until a persistent count was achieved. Emergence of the first seedling was considered as the time to start emergence (TSE). The formula proposed by Ellis and Roberts [39] was adopted to measure the mean emergence time:Mean emergence time (MET) = Ʃ Dn/Ʃ n
where D is the number of days counted from the beginning of emergence, and n is the number of seeds which emerged on the respective day.

To determine the emergence index, the formula proposed by the Association of Official Seed Analysis [40] was followed:Emergence index (EI) = (number of emerged seedling(s)/day of first count) + - - - + (number of emerged seedlings/idem of final count)

The final emergence percentage was calculated according to following formula:Final emergence percentage (FEP) = number of emerged seedlings/number of seeds sown × 100

### 2.4. Measurement of Growth Attributes

Seedlings were harvested 30 days after emergence from each experimental unit to record the data of seedling growth. A meter rod was used to measure the lengths of shoots and roots. An electronic weighing balance was used to record the data of the fresh and dry biomasses of rice seedlings. After recording the fresh biomass, to record the dry biomass, seedlings were placed in the oven at 70 °C until a constant weight was achieved.

### 2.5. Estimation of Physiological Attributes

At the harvesting time (30 days after emergence), samples were collected from the flag leaf of mother tillers from each experimental unit to record the data regarding chlorophyll pigments (chlorophyll *a* and *b* and carotenoid contents). A spectrophotometer was used to analyze the chlorophyll pigments of the leaf samples of rice seedlings. In accordance with Arnon [41], absorbance of filtrates at 663, 645 and 480 nm for chlorophyll *a* and *b* and carotenoids, respectively, was observed, and the following formulas were used to measure the chlorophyll pigment concentrations in the leaf samples:
Chlorophyll *a* content = [12.7 (OD 663) − 2.69 (OD 645)] × V/1000 × W
Chlorophyll *b* content = [12.7 (OD 645) − 4.68 (OD 663)] × V/1000 × W
Carotenoids = [(OD 480) + 0.114 (OD 663) − 0.638 (OD 645)] × V/1000 × W
where
V = volume of extract in mL;W = weight of sample (fresh leaf) in grams (g).

By summation of the chlorophyll a content and chlorophyll b content, the total chlorophyll content was recorded: Total chlorophyll content = chlorophyll *a* content + chlorophyll *b* content

### 2.6. Measurement of Gas Exchange Attributes

An infrared gas analyzer (IRGA LI-6400; a portable device) was used for the estimation of the photosynthesis rate (*A*; µmol CO_2_ m^−2^ s^−1^), stomatal conductance (*gs*; mmol m^−2^ s^−1^) and respiration rate (*E*; mmol H_2_O m^−2^ s^−1^). The flag leaf of mother tillers from each experimental unit was considered suitable to record the data of gas exchange attributes. Data of gas exchange attributes were recorded for two hours between 10:00 and 12:00 a.m. by using the flag leaf of rice seedlings according to procedures described by Long [42].

### 2.7. Determination of Enzymatic Activities

Samples from the flag leaf of mother tillers of each experimental unit were collected at the harvesting time to estimate the enzymatic activities. The superoxide dismutase activity was determined according to the method described by Giannopolitis and Ries [43]. The procedure proposed by Chance and Maehly [44] was followed to estimate the activity of catalase. With a slight modification, the protocol suggested by Nakano and Asada [45] for ascorbate peroxidase was followed accordingly to observe the ascorbate peroxidase activity. The method proposed by Velikova [46] was used to measure the concentration of hydrogen peroxide.

### 2.8. Statistical Analysis

Collected data of emergence, physiological and gas exchange attributes, growth and enzymatic activities were analyzed and evaluated statistically by using the statistical package “Statistic 8.1”, employing Fisher’s analysis of variance (ANOVA) technique under a completely randomized design. Microsoft Excel was used for calculation and graphical presentation. The statistical model includes the replications (4), drought levels (2), foliar treatments (5) of MLEs and the interaction between drought levels and foliar treatments of MLEs. Different letters (a, b, c, etc.) were used to portray the significant differences among treatments’ effects via LSD at a 5% probability level [47].

## 3. Results

The effect of the moringa leaf extracts as a priming agent as well as the water regimes on rice seedlings was studied through various indicators such as emergence and growth attributes, photosynthetic pigments, gas exchange parameters and antioxidant activities. Significant levels of seedling emergence and growth, photosynthetic pigments, gas exchange parameters and antioxidant activities, as influenced by priming agents and water regimes, are presented in Table 1.

### 3.1. Emergence Attributes

The time to start emergence, mean emergence time and emergence index of rice seedlings were significantly influenced by the priming agents and water regimes, but the interaction was statistically non-significant (Table 1). A lower time to start emergence (5.15 days) and mean emergence time (10.74 days) of seedlings were recorded under well-watered conditions as compared to the water deficit regime (Table 2). The water deficit regime increased the time to start emergence (5.65 days) and mean emergence time (10.91). In the case of priming agents, the maximum reduction in the time to start emergence was observed for FM priming, which was statistically at par with DM and EM priming. FM priming also reduced the mean emergence time of rice seedlings (10.60 days). FM priming and the normal regime of water treatment produced the maximum emergence index (6.97). Water deficit conditions also reduced the emergence index (5.13) of seedlings. The final emergence percentage was statistically improved by all priming agents, but the maximum outcome was recorded for FM priming (98.75%) (Table 2). The water regimes did not significantly influence the final emergence percentage.

### 3.2. Growth Attributes

Growth attributes including the fresh and dry biomasses and the shoot and root lengths of rice seedlings were significantly affected by the priming agents and water treatments (Table 1), but the interaction was only significant regarding the dry biomass of rice seedlings. All the priming agents enhanced the fresh and dry biomasses and the shoot and root lengths of rice seedlings, but the maximum improvement was observed for FM priming (Table 3). The water deficit regime reduced the fresh and dry biomass as well as shoot length of seedlings, while root length was increased. The maximum dry biomass (14.43 g) was recorded for FM priming under the control water treatment, while the minimum (8.45 g) was recorded under the water deficit regimes (Table 3).

### 3.3. Photosynthetic Pigments 

Chlorophyll *a* and *b* and total chlorophyll contents were significantly affected by the priming treatments as well as water regimes, and even the interaction of both factors was significant (Table 1), while the interaction was non-significant regarding carotenoids. FM priming was responsible for the maximum synthesis of chlorophyll contents including chlorophyll *a* and *b*, total chlorophyll and carotenoid contents (Figure 1). FM priming performed better under water deficit regimes as well. FM priming produced the maximum total chlorophyll contents under control conditions followed by the same treatment under water deficit regimes (Figure 1). DM and EM priming were statistically at par with each other regarding chlorophyll pigments. A lower concentration of chlorophyll pigments was observed in non-primed seeds as well as in hydro priming. Water regimes also influenced the concentration of photosynthetic pigments in rice seedlings. Water deficit regimes reduced the production of photosynthetic pigments, while the maximum concentration was recorded under normal water regimes (Figure 1).

### 3.4. Gas Exchange Attributes

The priming agents and water regimes significantly influenced the gas exchange attributes, i.e., photosynthesis rate (*A*), stomatal conductance to water (*gs*) and respiration rate (*E*), of rice seedlings, while their interaction was non-significant (Table 1). A better photosynthesis rate was observed under control conditions as compared to drought stress. In the case of priming, all the priming agents showed an improvement in the photosynthesis rate in normal control conditions as well as under water deficit regimes, but the maximum response was recorded by FM priming, followed by DM and EM priming, in that order (Figure 2). The minimum photosynthesis rate was observed in control and hydro priming treatments. FM priming also improved the stomatal conductance under the water deficit regime, which was statistically similar under the normal condition. The water deficit regime reduced the respiration rate as compared to the normal condition, while FM priming improved the respiration rate either in normal or deficit conditions (Figure 2). The minimum respiration rate was found under non-primed and hydro priming treatments, which were statistically at par with each other.

### 3.5. Antioxidant Activities

The activities of superoxide dismutase, catalase, ascorbate peroxidase and H_2_O_2_ were significantly affected by the water regimes and priming agents (Table 1). The water deficit regimes improved the antioxidant activities of rice seedlings, while low antioxidant activities were observed under normal conditions (Figure 3). Drought stress significantly enhanced the activities of superoxide dismutase, catalase, ascorbate peroxidase and H_2_O_2_ comparatively. All priming agents, particularly FM priming, improved the activities of superoxide dismutase, catalase and ascorbate peroxidase under normal and water deficit regimes, but the maximum improvement was recorded by FM priming. On the other hand, the H_2_O_2_ concentration was increased by water deficit regimes, but the priming agents reduced the concentration of H_2_O_2_ either under normal or water deficit regimes; the maximum reduction was noted by FM priming under both water regimes (Figure 3). DM and EM priming performed statistically similarly under the normal and deficit regimes. The maximum production of H_2_O_2_ was recorded in non-primed and hydro priming treatments under water deficit regimes (Figure 3).

## 4. Discussion

In this study, it was observed that water regimes and priming agents significantly influenced the emergence and growth attributes, photosynthetic pigments, gas exchange parameters and antioxidant activities. Under drought stress, plants showed a reduction in the concentrations of photosynthetic pigments and rate of gas exchange attributes, but with enhanced enzymatic activities, in the present experimentation. The priming agents appeared to mitigate the adverse impacts of water deficit by improving the emergence and growth attributes of rice seedlings, as also shown in wheat seeds [26]. The improved emergence attributes (time to start emergence, mean emergence time, emergence index and final emergence percentage) of rice seedlings with respect to the controls may be associated with greater and more rapid intake of water by the seeds (imbibition process) and the triggering of enzymatic activities linked with rupturing of the seed coat due to the seed priming agent [30,48]. The improved speed and spread of emergence indicate the considerable potential of MLEs for agricultural use as priming agents to improve drought tolerance and thereby increase crop productivity in environmentally stressed regions. Our scientific outcomes are also supported by previous studies demonstrating that plant growth and vigor were improved by application of MLEs as priming agents in maize [33]. Our findings are also supported by Khan et al. [34], who found that a leaf extract from a moringa landrace from the Faisalabad region had more growth-enhancing potential as compared to the leaf extracts of other available landraces regarding seedling growth and vigor. Farooq and Koul [49] observed variation regarding various bioactive compounds in the seed and leaf extracts of various cultivars of moringa. They concluded that seed and leaf extracts of the Jaffin variety contain strong antioxidant and antibacterial activities with greater total flavonoid and phenolic contents as compared to other landraces. Therefore, higher plant growth-enhancing activity was present in the extract of the Jaffin cultivar. Seed priming enhances the rate of metabolism, which results in an increase in the speed of germination and emergence [50].

Growth parameters including the fresh and dry biomasses and the shoot and root lengths of rice seedlings were adversely affected by the water deficit regime in the present experimentation. On the other hand, application of MLEs as priming agents enhanced the growth of rice seedlings under normal conditions, but improvement was also observed under the water deficit regime. Not only are moringa leaves good reservoirs of mineral nutrients, i.e., K, Ca, Mg and vitamin C, but they also contain zeatin, a type of cytokinin, which speeds up the cell division process [28]. These observations suggest that the mechanism of improved growth by MLE application is based on the improvement in emergence attributes of seedlings and the subsequent increase in the plant growth components. Thus, the application of MLEs as priming agents may enhance the performance of rice seedlings. Most of the mineral nutrients of MLEs during seed priming seemed to be partitioned to the embryo of the seed, which boosted the emergence of seedlings and ultimately the growth and development of plants. Seed priming can be particularly beneficial to resource-poor farmers working in low-input agricultural systems where the yield potential is limited by intrinsically stressed agronomic environments [32]. The findings of the current study are also in line with the outcomes of Khan et al. [29,34], who found that extracts prepared from the leaves of landraces in Faisalabad exhibited a greater biostimulant potential that might be due to the availability of bioactive compounds, antioxidants, mineral nutrients and plant growth-promoting substances. Moreover, Basra et al. [33] reported that seed priming of maize with MLEs improved plant growth and economic yield. Similarly, Abdalla [51] noted that a foliar spray of MLEs enhanced the plant length, fresh (68.1%) and dry biomasses (51.5%), photosynthetic pigments, photosynthetic rate, stomatal conductance, total soluble protein, ascorbic acid content and phytohormones in rocket plants.

In the current study, MLE treatment significantly improved the chlorophyll pigments and gas exchange attributes under water shortage. Production of chlorophyll pigments improves efficiency and utilization of inputs. Water use efficiency is also improved by higher production of chlorophyll pigments. As the concentration of chlorophyll pigments increases, gas exchange attributes are also improved. Photosynthetic and respiration activities are enhanced, which are responsible for the stay-green period. MLEs are rich in cytokinins which delay leaf senescence, resulting in a high leaf area and a greater amount of chlorophyll pigments [52]. It has been previously reported that exogenous application of MLEs improved the chlorophyll pigments [28], gas exchange attributes, antioxidant activities, growth and yield of field crops cultivated under normal and stressful circumstances [53,54]. Similar findings were also noted in rocket plants, where a foliar spray of MLEs enhanced the plant length, fresh and dry masses, chlorophyll contents, photosynthetic rate, stomatal conductance, ascorbic acid, total soluble proteins and phytohormones [51]. Moringa leaves are an excellent source of zeatin (5 to 200 µg/g of fresh leaves), antioxidants and tocopherols which enhance plant tolerance against environmental stresses [55,56]. The findings of the current experiment are also in line with a previous study in which MLE use increased the chlorophyll contents in abiotically stressed wheat. In addition, exogenous use of MLEs improved the wheat chlorophyll pigments [13]. Improved plant growth with fresh MLEs can be attributed to the presence of various secondary metabolites such as phenols and ascorbates [33]. These findings are also supported by Khan et al. [34], who found that a moringa leaf extract from a landrace of Faisalabad origin showed a higher biostimulant potential that might be due to the presence of a high concentration of plant growth-promoting compounds, mineral nutrients and antioxidants. Use of MLEs from all landraces significantly enhanced the morphological, physiological and yield traits both under ambient and stressed conditions as compared to the controls [29]. The application of minerals, either alone or in combination with growth promoters, improved the growth attributes of field crops [57].

In biochemical attributes, MLE application showed positive synergy with other bio-chemical stimulants. In the present findings, the improvement in biochemical attributes may be due to the presence of allelochemicals and secondary compounds such as ascorbic acid, total phenolics [28] and zeatin [58]. Drought stress adversely reduced the chlorophyll and carotenoid contents, and their reduction was directly associated with the intensity of stress. In the current study, drought stress decreased the photosynthetic pigments, which is in line with previous findings in which water shortage lowered the chlorophyll and carotenoid contents of rice [59]. Drought stress decreased the level of chlorophyll contents, while under ambient conditions, more chlorophyll pigments were observed as compared to the drought stress condition [60]. Zhang et al. [61] also reported that drought stress reduced the photosynthetic rate, water contents and transpiration rate and enhanced stomatal resistance. Thus, the chlorophyll content and photosynthetic rate can be used to check the intensity of drought stress [62]. The application of mineral elements and organic compounds is considered a helpful practice in maintaining crop productivity with improved soil fertility to achieve the maximum plant growth and economical yield under stressful conditions [63,64].

Antioxidants play an inevitable role in increasing plant tolerance against abiotic stress by improving the plant defense system. In the current study, exogenous use of MLEs increased the antioxidant activities (SOD, CAT and APX), which protect the plants from oxidative damage. The activities of antioxidants are higher under stressful conditions as compared to normal and favorable conditions, which protects cells and organelles from oxidative damage [65]. Therefore, to increase plant tolerance, there is a need to improve antioxidant activity to overcome oxidative stress. SOD acts as the basis of defense in response to oxidative stress [66]. SOD expression plays a vital role against drought stress by scavenging the H_2_O_2_ concentration [67]. Hanafy [68] found that under drought, the activity of the enzymatic antioxidants glutathione reductase (GR), SOD and APX was enhanced in soybean and canola [69]. Moreover, in common bean, MLE treatment, both foliar spray and seed priming, significantly increased enzymatic antioxidant activities such as those of GR, SOD and APX [70]. Moringa leaf extract (MLE) is a natural seed priming agent and crop growth enhancer. The farmer community can use MLE at 3% as a seed priming agent to enhance the growth and productivity of field crops cultivated under normal and/or unfavorable conditions. 

## 5. Conclusions

The results obtained from the experimentation show that the water deficit regime adversely affected the studied indicators including emergence and growth attributes as well as physiological parameters. Among the priming agents, the MLE from the Faisalabad landrace significantly improved the speed and spread of emergence of rice seedlings. Therefore, the MLE of the Faisalabad landrace can be productively used to boost the stand establishment of seedlings and growth of rice grown under normal and water deficit conditions. Further research is required to support MLEs’ impact as seed priming agents on the final yield, even under field conditions.

## Figures and Tables

**Figure 1 plants-11-00261-f001:**
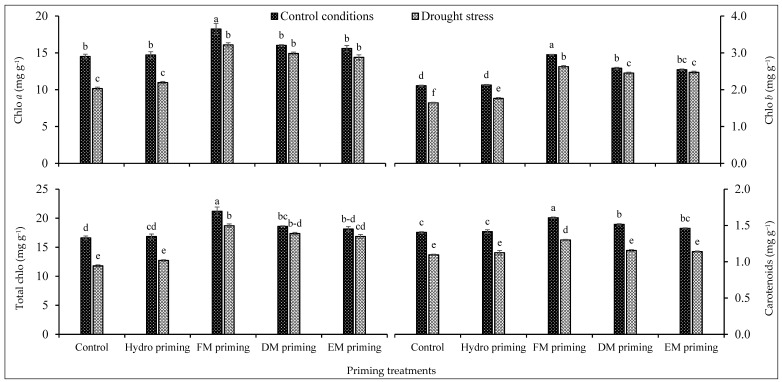
Impact of moringa leaf extracts as priming agents on photosynthetic pigments of rice seedlings cultivated under control and water deficit regimes. Bars sharing the same letter did not differ significantly at *p* = 0.05.

**Figure 2 plants-11-00261-f002:**
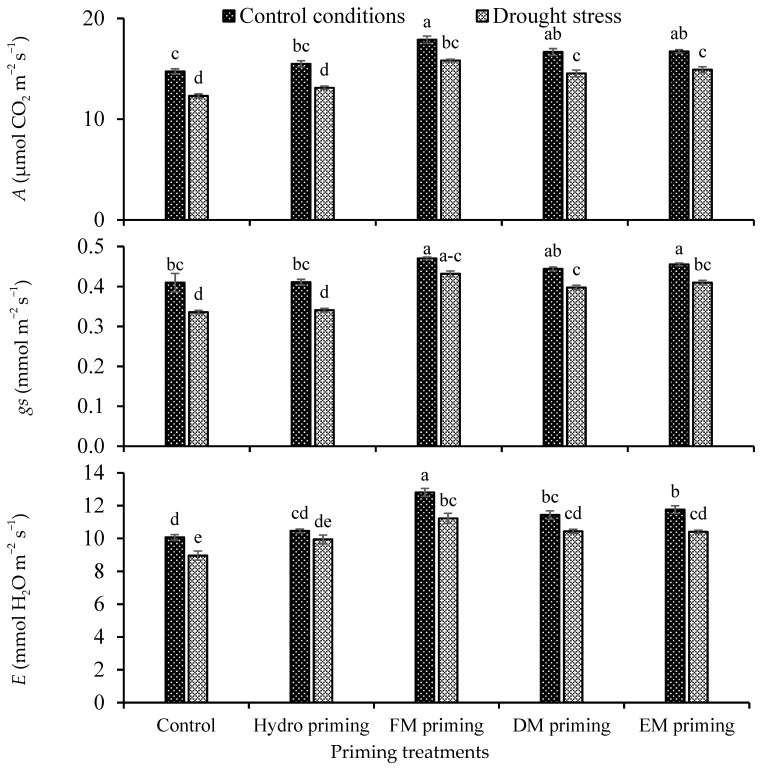
Impact of moringa leaf extracts as priming agents on gas exchange attributes of rice seedlings cultivated under control and water deficit regimes. Bars sharing the same letter did not differ significantly at *p* = 0.05.

**Figure 3 plants-11-00261-f003:**
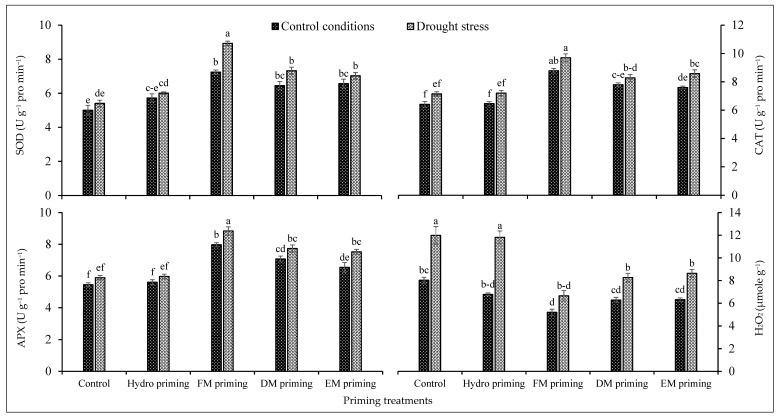
Impact of moringa leaf extracts as priming agents on antioxidant activities of rice seedlings cultivated under control and water deficit regimes. Bars sharing the same letter did not differ significantly at *p* = 0.05.

**Table 1 plants-11-00261-t001:** Analysis of variance for emergence, growth, photosynthetic pigment, gas exchange and enzymatic attributes of rice seedlings cultivated under normal and water deficit regimes in response to priming treatments of moringa leaf extracts (n = 4).

SOV	DF	TSE	MET	EI	FEP	Fresh Biomass	Dry Biomass	Shoot Length	Root Length	Chlo *a*	Chlo *b*
WT	1	2.500 **	0.306 *	3.283 **	122.50NS	1343 **	142.8 **	1648 **	262 **	63.17 **	0.739 **
PT	4	1.712 **	0.155 *	10.437 **	1433.75 **	145.4 **	43.09 **	255 **	179 **	31.46 **	1.260 **
WT × PT	4	0.062 ^NS^	0.036 ^NS^	0.145 ^NS^	16.25 ^NS^	1.22 ^NS^	1.698 **	3.87 ^NS^	2.15 ^NS^	4.38 **	0.053 **
SOV	DF	Total Chlo	Carotenoids	*A*	*gs*	*E*	SOD	CAT	APX	H_2_O_2_	
WT	1	77.59 **	1.004 **	46.91 **	0.0297 **	12.321 **	5.453 **	5.952 **	4.389 **	87.26 **	
PT	4	45.18 **	0.051 **	13.78 **	0.0098 **	7.136 **	9.594 **	8.386 **	10.521 **	21.65 **	
WT × PT	4	5.34 **	0.001 ^NS^	0.129 ^NS^	0.0005 ^NS^	0.325 ^NS^	0.660 *	0.075 ^NS^	0.142 ^NS^	4.459 **	

SOV = source of variance, WT = water treatment, PT = priming treatment, WT × PT = interaction of water and priming treatments, DF = degrees of freedom, NS = non-significant, TSE = time to start emergence, MET = mean emergence time, EI = emergence index, FEP = final emergence percentage, Chlo = chlorophyll, *A* = photosynthesis rate, *gs* = stomatal conductance to water, *E* = respiration rate, SOD = superoxide dismutase, CAT = catalase, APX = ascorbate peroxidase, * = significant at *p* < 0.05, ** = significant at *p* < 0.01.

**Table 2 plants-11-00261-t002:** Impact of moringa leaf extracts from various landraces as priming agents on emergence attributes of rice seedlings cultivated under control and water deficit regimes.

Treatments	Time to Start Emergence (Days)			Mean Emergence Time (Days)			Emergence Index			Final Emergence Percentage (%)		
	CC	DS	Mean (PT)	CC	DS	Mean (PT)	CC	DS	Mean (PT)	CC	DS	Mean (PT)
Control	5.75	6.25	6 A	10.91	11.05	10.98 A	4.26	3.69	3.97 C	70	65	67.5 B
Hydro priming	5.5	6	5.75 AB	10.87	10.87	10.87 AB	4.92	4.48	4.70 C	77.5	70	73.75 B
FM priming	4.5	5.25	4.875 C	10.55	10.65	10.60 B	7.07	6.87	6.97 A	100	97.5	98.75 A
DM priming	5	5.25	5.125 BC	10.65	10.99	10.82 AB	6.27	5.45	5.86 B	92.5	92.5	92.5 A
EM priming	5	5.5	5.25 A-C	10.73	11.01	10.87 AB	5.97	5.14	5.56 B	92.5	90	91.25 A
Mean (WT)	5.15 B	5.65 A		10.74 B	10.91 A		5.70 A	5.13 B		86.5 A	83 A	
HSD	PT = 0.764, WT = 0.339, PT × WT = NS			PT = 0.322, WT = 0.1430, PT × WT = NS			PT = 0.784, WT = 0.347, PT × WT = NS			PT = 10.76, WT = NS, PT × WT = NS		

Means sharing the same letters in the same column did not differ significantly at *p* ≤ 0.05. NS = statistically non-significant, CC = control conditions, DS = drought stress, PT = priming treatment, WT = water treatment, FM = moringa leaf extract from local landrace of Faisalabad origin, DM = Dera Ghazi Khan origin moringa, EL = exotic landrace of moringa. Means sharing the same letter did not differ significantly at *p* = 0.05.

**Table 3 plants-11-00261-t003:** Impact of moringa leaf extracts from various landraces as priming agents on growth attributes of rice seedlings cultivated under control and water deficit regimes.

Treatments	Fresh Biomass (g)			Dry Biomass (g)			Shoot Length (cm)			Root Length (cm)		
	CC	DS	Mean (PT)	CC	DS	Mean (PT)	CC	DS	Mean (PT)	CC	DS	Mean (PT)
Control	31.26	19.61	25.43 C	10.56 d	6.99 f	8.85 C	35.66	20.45	28.05 C	12.41	18.90	15.66 D
Hydro priming	32.33	19.53	25.93 C	11.05 cd	7.08 ef	9.02 BC	36.84	24.00	30.42 C	16.87	22.12	19.49 C
FM priming	41.75	30.38	36.06 A	16.34 a	12.52 b	14.43 A	48.31	35.87	42.09 A	25.84	31.26	28.55 A
DM priming	35.19	23.70	29.44 B	12.14 bc	7.18 ef	9.61 BC	43.20	31.42	37.31 B	20.12	24.94	22.53 B
EM priming	35.43	24.79	30.11 B	11.05 cd	8.51 e	9.78 B	42.71	30.78	36.74 B	20.75	24.38	22.57 B
Mean (WT)	35.19 A	23.60 B		12.23 A	8.45 B		41.34 A	28.50 B		19.20 B	24.32 A	
HSD	PT = 1.259, WT = 0.558, PT × WT = NS			PT = 0.874, WT = 0.387, PT × WT = 1.455			PT = 1.871, WT = 0.829, PT × WT = NS			PT = 1.484, WT = 0.658, PT × WT = NS		

Means sharing the same letters did not differ significantly at *p* ≤ 0.05. NS = statistically non-significant, CC = control conditions, DS = drought stress, PT = priming treatment, WT = water treatment, FM = moringa leaf extract from local landrace of Faisalabad origin, DM = Dera Ghazi Khan origin moringa, EL = exotic landrace of moringa. Means sharing the same letter did not differ significantly at *p* = 0.05.

## Data Availability

The data that support the outcomes of the current experimentation are available from the corresponding author (S.K.) upon reasonable request.

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
