# Peer review of "Application of Moringa Leaf Extract as a Seed Priming Agent Enhances Growth and Physiological Attributes of Rice Seedlings Cultivated under Water Deficit Regime"

_plants, 2022, doi:10.3390/plants11030261_

Round 1
Reviewer 1 Report
I have been already finished to evaluate the manuscript entitled "Application of moringa leaf extract as a seed priming agent enhances growth and physiological attributes of rice seedlings cultivated under water deficit regime". It is a good idea to seed priming with moringa leaf extract. By the way, I have some major issues to be concerned as:
- What is a hypothesis on long term effects of seed priming with biostimulants?
- Only seed germination data are understanding as seed priming strategies. How about the long term function as biostimulant?
- In present manuscript, 1) seed gemination and 2) flag leaf data are presented. How about the overall growth performances and yield traits? Rice grain is one of the most traits to be reported.
- In the previous publication of the authors (year 2021 in PLoS ONE), the exogenous foliar-application in rice crop using moringa leaf extract has been done. What is a different mode of action in this compound? Why do you select only 75% FC as drought stress in present study?
- What is a novelty of this report?
Also, there are several minor issues to be edited. For example,
- Scientific name of each organism would be represented as italic font. Please check one by one.
- Please check "Factor B; Foliar treatments". Is seed priming as treatments?
- Please check "75% field capacity" as moderate drought stress. Why do you select as it is?
- Are there any chemical profiles of moringa water extraction? What is the dominated compounds and function?
- "gs" and "Gs" Please make a uniform abbreviation.
- Are the authors collect the data of chlorophyll fluorescence?
- How about the lipid peroxidation and membrane leakage in relation to antioxidant activities?
- The reference list would be checked and edited one-by-one, especially journal abbreviation.
This is my opinion.
Author Response
Subject: Response to Comments
Title: Application of moringa leaf extract as a seed priming agent enhances growth and physiological attributes of rice seedlings cultivated under water deficit regime.
We are very much thankful to reviewer for sparing time to review our manuscript and provided comments and suggestions for the improvement of the manuscript.
I, Shahbaz Khan, corresponding author of the manuscript, am enclosing herewith a revised manuscript entitled “Application of moringa leaf extract as a seed priming agent enhances growth and physiological attributes of rice seedlings cultivated under water deficit regime” for publication in “Plants” after possible improvements. All the comments and suggestions are addressed accordingly and incorporated in the revised manuscript. Details of individual comments are given below.
REVIEWER 1
General Comments
I have been already finished to evaluate the manuscript entitled "Application of moringa leaf extract as a seed priming agent enhances growth and physiological attributes of rice seedlings cultivated under water deficit regime". It is a good idea to seed priming with moringa leaf extract. By the way, I have some major issues to be concerned as:
Response: Thank you so much for your time for reviewing our manuscript and providing comments and suggestions to improve the quality of manuscript. All the comments and suggestions are incorporated and highlighted in the revised manuscript.
Comment: What is a hypothesis on long term effects of seed priming with biostimulants?
Response: Generally, seed priming improves the speed and spread of emergence/germination. Healthy seedlings are produced by seed priming with biostimulant which are responsible for good stand establishment of seedling. Uniform and quick emergence ensures the heathy crop growth which ultimately improves the economical/grain yield.
Comment: Only seed germination data are understanding as seed priming strategies. How about the long term function as biostimulant?
Response: Seed priming improves the final emergence/germination percentage and plant population. Seed priming is not directly involved in seed enhancement but it plays important role indirectly by improving the emergence attributes and healthy seedlings.
Comment: In present manuscript, 1) seed germination and 2) flag leaf data are presented. How about the overall growth performances and yield traits? Rice grain is one of the most traits to be reported.
Response: The current study was designed to explore the impact of leaf extracts from various moringa landraces up to seedling stage. We will consider your suggestions and recommendations in future studies.
Comment: In the previous publication of the authors (year 2021 in PLoS ONE), the exogenous foliar-application in rice crop using moringa leaf extract has been done. What is a different mode of action in this compound? Why do you select only 75% FC as drought stress in present study?
Response: Moringa leaf extracts contains mineral elements and growth hormone like cytokinins (zeatin) which increase cell division and ultimately growth is improved. More levels of drought make the experimentation complicated, so only two levels of water regimes were under current study.
Comment: What is a novelty of this report?
Response: In this study, we compare the seed priming potential of leaf extract from three moringa landraces; white seeded moringa (local landrace, Faisalabad origin), black seeded moringa (local landrace, DG Khan origin) and PKM1 moringa (exotic landrace (India), established at Faisalabad). “Maximum seed priming potential was observed in leaf extract from white seeded moringa, local landrace of Faisalabad” is the novelty of current study.
Comment: Scientific name of each organism would be represented as italic font. Please check one by one.
Response: Suggestions are incorporated and highlighted.
Comment: Please check "Factor B; Foliar treatments". Is seed priming as treatments?
Response: Suggestions are incorporated and highlighted.
Comment: Please check "75% field capacity" as moderate drought stress. Why do you select as it is?
Response: There are only two levels water treatments (control conditions and drought stress). Soil moisture content at 100% was considered as control conditions while soil moisture content at 75% as drought stress. There was no severe and/or moderate drought stress. In future studies, we will consider more levels of water treatments.
Comment: Are there any chemical profiles of moringa water extraction? What is the dominated compounds and function?
Response: At present, there is no chemical profile of moringa water extraction. We will try to fulfil recommendation in future experiments.
Comment: "gs" and "Gs" Please make a uniform abbreviation.
Response: Suggestions are incorporated and highlighted.
Comment: Are the authors collect the data of chlorophyll fluorescence?
Response: Data of chlorophyll fluorescence were not recorded. We collected the fresh samples from flag leaf and analyze the chlorophyll pigments by using spectrophotometer.
Comment: How about the lipid peroxidation and membrane leakage in relation to antioxidant activities?
Response: In future studies, we will observe lipid peroxidation and membrane leakage in relation to antioxidant activities.
Comment: The reference list would be checked and edited one-by-one, especially journal abbreviation.
Response: Suggestions are incorporated and highlighted.
Reviewer 2 Report
This manuscripts makes an analysis of the impact of Moringa leaves extracts on the germination of rice.
The introduction and the problematic has been well descrived and supported by the litterature, but some references are missing (please see more details in the document attached)
The materials and methods are well described.
Results are clearly presented and analysed, only one remark about the form of figures, see detail on the document.
I've made some corrections and suggestions on the text.
I suggest to open more the conclusions to perspectives on the potential use of those extracts on fields if you can add somelines on it will be better.

Author Response
Response Sheet
Subject: Response to Comments
Title: Application of moringa leaf extract as a seed priming agent enhances growth and physiological attributes of rice seedlings cultivated under water deficit regime.
We are very much thankful to reviewer for sparing time to review our manuscript and provided comments and suggestions for the improvement of the manuscript.
I, Shahbaz Khan, corresponding author of the manuscript, am enclosing herewith a revised manuscript entitled “Application of moringa leaf extract as a seed priming agent enhances growth and physiological attributes of rice seedlings cultivated under water deficit regime” for publication in “Plants” after possible improvements. All the comments and suggestions are addressed accordingly and incorporated in the revised manuscript. Details of individual comments are given below.
REVIEWER 2
General Comments
This manuscript makes an analysis of the impact of Moringa leaves extracts on the germination of rice.
Comment: The introduction and the problematic has been well described and supported by the literature, but some references are missing (please see more details in the document attached).
Response: Suggestions are incorporated and highlighted.
Comment: The materials and methods are well described.
Response: Thank you for your comments.
Comment: Results are clearly presented and analysed, only one remark about the form of figures, see detail on the document.
Response: Suggestions are incorporated and highlighted.
Comment: I've made some corrections and suggestions on the text.
Response: Suggestions are incorporated and highlighted.
Comment: I suggest to open more the conclusions to perspectives on the potential use of those extracts on fields if you can add somelines on it will be better.
Response: Suggestions are incorporated and highlighted.
Furthermore, all the comments and suggestions given in the text file are incorporated and highlighted accordingly.
We are very much thankful to reviewer for giving the suggestions and comments to improve the manuscript.
Round 2
Reviewer 1 Report
Overall is accepted. By the way, please intensively check and correct the minor issues i.e. "H2O2", "gs", etc. In the reference list, please check an abbreviation of journal one-by-one. It needs to be corrected. This is my opinion.
Author Response
Dear Reviewer,
Thank you so much to review our manuscript and providing comments and suggestions for the improvement.
I have made all possible changes regarding "H2O2", "gs", etc. I also checked the reference list and made abbreviation of journal(s) one-by-one accordingly. These changes are also highlighted in the revised manuscript.
Thanks in anticipation.
Corresponding Author.